# Genetic Organization of Acquired Antimicrobial Resistance Genes and Detection of Resistance-Mediating Mutations in a *Gallibacterium anatis* Isolate from a Calf Suffering from a Respiratory Tract Infection

**DOI:** 10.3390/antibiotics12020294

**Published:** 2023-02-01

**Authors:** Anne-Kathrin Schink, Dennis Hanke, Torsten Semmler, Nicole Roschanski, Stefan Schwarz

**Affiliations:** 1Institute of Microbiology and Epizootics, Centre for Infection Medicine, Department of Veterinary Medicine, Freie Universität Berlin, 14163 Berlin, Germany; 2Veterinary Centre for Resistance Research (TZR), Department of Veterinary Medicine, Freie Universität Berlin, 14163 Berlin, Germany; 3NG1-Microbial Genomics, Robert Koch Institute, 13353 Berlin, Germany; 4Landeslabor Schleswig-Holstein, 24537 Neumünster, Germany

**Keywords:** *Gallibacterium anatis*, antimicrobial resistance, cattle, multidrug resistance, respiratory disease, whole-genome sequencing

## Abstract

*Gallibacterium* (G.) *anatis* isolates associated with respiratory diseases in calves and harboring acquired antimicrobial resistance genes have been described in Belgium. The aim of this study was to analyze the genetic organization of acquired resistance genes in the *G. anatis* isolate IMT49310 from a German calf suffering from a respiratory tract infection. The isolate was submitted to antimicrobial susceptibility testing, and a closed genome was obtained by a hybrid assembly of Illumina MiSeq short-reads and MinION long-reads. Isolate IMT49310 showed elevated MIC values for macrolides, aminoglycosides, florfenicol, tetracyclines, and trimethoprim/sulfamethoxazole. The acquired resistance genes *catA1*, *floR*, *aadA1*, *aadB*, *aphA1*, *strA*, *tet*(M), *tet*(B), *erm*(B), and *sul2* were identified within three resistance gene regions in the genome, some of which were associated with IS elements, such as IS*Vsa5*-like or IS*15DII*. Furthermore, nucleotide exchanges within the QRDRs of *gyrA* and *parC*, resulting in amino acid exchanges S83F and D87A in GyrA and S80I in ParC, were identified. Even if the role in the pathogenesis of respiratory tract infections in cattle needs to be further investigated, the identification of a *G. anatis* isolate with reduced susceptibility to regularly used antimicrobial agents in cases of fatal bovine respiratory tract infections is worrisome, and such isolates might also act as a reservoir for antimicrobial resistance genes.

## 1. Introduction

*Gallibacterium* (G.) *anatis* is a Gram-negative bacterium that belongs to the family Pasteurellaceae. It is known as a commensal inhabitant of the respiratory, intestinal, and genital tract in poultry but has also been described as an opportunistic pathogen involved in the pathogenesis of respiratory tract infections, such as tracheitis or aerosacculitis, and genital tract infections, such as salpingitis and oophoritis [1]. Moreover, *G. anatis* has been associated with systemic infections, including pericarditis, perihepatitis, peritonitis, and septicemia in poultry, and studies reported high percentages of isolates showing elevated minimal inhibitory concentrations (MICs) for different classes of antimicrobial agents, but information about acquired antimicrobial resistance genes remains scarce [1,2,3,4,5]. *G. anatis* has also been isolated from infection sites in mammalian hosts, including humans and cattle [1,6,7]. Multifactorial respiratory tract infections are common in cattle, especially among calves, when transferred between sectors of beef cattle production and account for significant financial losses and reduced animal health [8]. In the pathogenesis of bovine respiratory tract diseases, bacteria such as *Pasteurella multocida*, *Mannheimia haemolytica*, and *Histophilus somni* are considered secondary pathogens, and treatment requires the application of antimicrobial agents [9]. Multidrug-resistant isolates of bovine respiratory pathogens have been identified, and the resistance genes corresponding to the resistance phenotypes were associated with mobile genetic elements (MGEs), such as insertion sequences (IS), transposons, plasmids, or integrative and conjugative elements (ICEs) [10]. In Belgium, *G. anatis* isolates displaying high MICs for tylosin, tetracycline, spectinomycin, kanamycin, and enrofloxacin have been isolated recently from calves suffering from unresponsive respiratory tract infections. This observation might point towards the involvement of *G. anatis* as an opportunistic pathogen in cattle and limited therapeutic options due to the detection of a multitude of acquired resistance genes in the sequences of *G. anatis* isolates [7]. The aims of this study were to identify which acquired antimicrobial resistance genes and resistance-mediating mutations are present as well as to gain insight into their genetic organization in the genome of a *G. anatis* isolate from a German calf suffering from a respiratory tract infection.

## 2. Results and Discussion

### 2.1. Antimicrobial Susceptibility Testing

As no interpretive criteria are currently available for *G. anatis*, a classification of the isolate as resistant or susceptible is not possible [11]. Nevertheless, antimicrobial susceptibility testing of *G. anatis* IMT49310 revealed elevated MIC values for several classes of antimicrobial agents. In particular, it showed high MIC values for the macrolides erythromycin (≥64 mg/L), tylosin (≥256 mg/L), tilmicosin (≥256 mg/L), and tulathromycin (32 mg/L), the lincosamides clindamycin (32 mg/L) and pirlimycin (≥128 mg/L), the aminoglycosides gentamicin (32 mg/L), streptomycin (32 mg/L), neomycin (8 mg/L), and kanamycin (≥128 mg/L), the phenicol florfenicol (32 mg/L), the tetracyclines tetracycline (64 mg/L) and doxycycline (16 mg/L), the (fluoro)quinolones enrofloxacin (16 mg/L), marbofloxacin (8 mg/L), ciprofloxacin (4 mg/L), and nalidixic acid (128 mg/L), as well as the combination trimethoprim/sulfamethoxazole (2/38 mg/L).

### 2.2. Sequence Analysis

Hybrid assembly of Illumina MiSeq short-reads and MinION nanopore long-reads resulted in a closed genome for *G. anatis* IMT49310. Sequence analysis revealed the presence of nucleotide substitutions resulting in amino acid exchanges within the quinolone resistance determining regions (QRDRs, *E. coli* numbering) of GyrA (S83F, D87A) and ParC (S80I). These substitutions have previously been described in Pasteurellaceae in conjunction with elevated MIC values for fluoroquinolones and might explain the high MIC values of *G. anatis* IMT49310 for ciprofloxacin, enrofloxacin, and marbofloxacin [7,10]. Eleven different acquired antimicrobial resistance genes were identified and found to be organized in three resistance gene regions.

The resistance gene region I (Figure 1A) was 8,236 bp in size (GenBank accession no. CP110225, bp 1,590,210–1,598,445) and contained the resistance genes *catA1* (chloramphenicol resistance), *aadA1* (streptomycin, spectinomycin resistance), and *aadB* (gentamicin, kanamycin resistance). The regions up and downstream of the resistance genes showed similarities to transposon Tn*As3* from *Aeromonas salmonicida* (GenBank accession no. CP000645). Upstream of the resistance genes, an integron-associated integrase gene of Tn*As3* was identified, followed by an open reading frame (*orf*) and the first 246 bp of a partial *tnpR* gene. Downstream of the resistance genes, the last 319 bp of a partial *tnpR* gene from Tn*As3*, consisting of 561 bp in sequence CP000645, was followed by a complete *tnpA* gene of Tn*As3*. Based on these structural comparisons (Figure 1A1), the resistance gene region I appeared to have been inserted in *orf*_01534, which could be found in the complete genome sequence of *G. anatis* strain UMN179 (GenBank accession no. CP002667 bp 1,662,354–1,663,655) (Figure 1A1). *G. anatis* strain UMN179 was isolated in Iowa (USA) from a laying hen suffering from peritonitis [12]. The resistance genes of *G. anatis* IMT49310 and the adjacent genes could also be found in different combinations in the sequences of other Gram-negative bacteria, either in the chromosomal DNA or on plasmids. For example, the chromosomal sequence of *E. coli* strain CFS3273 (GenBank accession no. CP026932) and the sequence of a plasmid (GenBank accession no. CP026933) from the same strain contained both the integrase and the *tnpA* genes of Tn*As3*, as well as *aadA1* and *aadB* in CP026932 and *catA1* in CP026933 (Figure 1A2). This shows that fragments of Tn*As3*, which were associated with resistance genes and located in different genetic settings, might have been involved in the formation of the resistance gene region I in *G. anatis* IMT49310.

The resistance gene region II (Figure 1B) of *G. anatis* IMT49310 had a size of 12,724 bp (GenBank accession no. CP110225, bp 1,610,266–1,622,989) and was located 121,273 bp downstream of the resistance gene region I. Resistance gene region II comprised the resistance genes *tet*(M) (tetracycline resistance) and two copies of *erm*(B) (macrolide and lincosamide resistance) and appeared to be inserted into *orf*_01557 (GenBank accession no. CP002667, bp 1,679,749–1,681,404) which is present in *G. anatis* UMN179. The resistance genes and the sequences immediately up and downstream showed homology to a Tn*916*-type integrative and conjugative element (ICE) from *Streptococcus pneumoniae* (GenBank accession no. FR671418). Between the two *erm*(B) copies, an *orf* showing homology to *orf*_10330 of *Jeotgalibaca ciconiae* (GenBank accession no. CP034465, bp 2,219,892–2,220,743) was identified (Figure 1B). The last three *orfs* of resistance gene region II in *G. anatis* IMT49310 aligned in the database with sequences from *Enterococcus faecalis* and other Gram-positive bacteria. In contrast to resistance gene region I, resistance gene region II was composed of genes mainly identified in Gram-positive bacteria, especially the segments containing the resistance genes *erm*(B) and *tet*(M) of the Tn*916*-type ICE. This observation underlines the ability of *G. anatis* to incorporate foreign DNA into its chromosome [13]. In this context, *G. anatis* might act as a mixing vessel for antimicrobial resistance genes from Gram-negative and Gram-positive bacteria.

The resistance gene region III (Figure 1C) was located 377,347 bp downstream of resistance gene region II and appeared to be inserted into the terminal part of *orf*_01265 identified at positions 1,368,549–1,369,589 bp in the *G. anatis* UMN179 sequence. The resistance gene region III was 13,216 bp (CP110225, bp 2,000,337–2,013,552) in size and flanked by IS*Vsa5*-like elements in opposite orientations. It contained the resistance genes *sul2* (sulfonamide resistance), *floR* (chloramphenicol, florfenicol resistance), *aadB* (gentamicin, kanamycin resistance), *aphA1* (kanamycin, neomycin resistance), *strA* (streptomycin resistance), and *tet*(B) (tetracycline resistance) (Figure 1C1). Similarities to sequences of different Gram-negative respiratory tract pathogens could be found in the database, such as *Pasteurella multocida* (GenBank accession no. CP029322, bp 1,843,468–1,849,967) and *Haemophilus parainfluenzae* (GenBank accession no. MW391932, bp 4,133–8,652). Moreover, parts of resistance gene region III aligned to different non-Pasteurellaceae species, e.g., to several different sections of *Proteus columbae* strain T60 (GenBank accession no. CP043925), including insertion sequences IS*Vsa5* and IS*15DII* (Figure 1C1,C2). As for the other two resistance gene regions, parts of resistance gene region III could be found in the database, but not the complete arrangement as seen in the *G. anatis* IMT49310 sequence. This region consisted of genes mainly identified in Gram-negative bacteria, and the insertion sequences might have played a role either in its formation or acquisition by *G. anatis* IMT49310. Neither of the flanking IS*Vsa5*-like elements exhibited the typical 9 bp direct repeats, but the 9 bp upstream the first IS*Vsa5*-like element (5′-CTGTAAGTA-3′) and the 9 bp downstream the second IS*Vsa5*-like element (5′-CTGTAAGTG-3′) closely resemble direct repeats, despite one nucleotide exchange (Figure 1C2). This might suggest that the entire resistance gene region III represents a composite transposon which, at its integration site in the *G. anatis* IMT49310 genome, produced imperfect direct repeats. As direct repeats occur as a repair mechanism of filling in single-stranded overhangs after insertion of the transposon, the observed single nucleotide exchange might have occurred accidentally. Furthermore, the insertion sequences might facilitate future mobilization and distribution of the resistance genes.

The resistance genes in the other two resistance gene regions were identified close to transposases, but large parts of the transposons to which they originally belonged were missing in *G. anatis* IMT49310. Therefore, the question remains whether the transposases were involved in the mobilization and acquisition of the resistance genes or in recombination events. Moreover, it is not clearly evident if the mobilization of resistance gene regions I or II as a whole would be facilitated. This is in contrast to other pathogens of the bovine respiratory tract. In *P. multocida* as well as *M. haemolytica* from the respiratory tract of a German calf, antimicrobial resistance genes were located in resistance gene regions in ICEs [14]. The resistance genes *floR*, *strA*, and *sul2* were among others present in ICE Tn*7406* from *M. haemolytica*, while Tn*7407* from *P. multocida* shared *aphA1*, *strA*, and *sul2* with *G. anatis* IMT49310. The resistance gene content showed some overlap between these Pasteurellaceae, but the organization of the genes differed distinctly, and translocation of one of these MGEs would transfer all contained antimicrobial resistance genes in one event, which would not be possible for the resistance genes in *G. anatis* IMT49310.

The resistance genes in *G. anatis* IMT49310 were also identified in *G. anatis* isolates from calves in Belgium, but no resistance genotype of the isolates matched exactly [7]. The resistance genes *aadA1*, *aadB*, and *catA1* also clustered on one contig and were associated with a Tn*As3* transposase as in resistance gene region I, but as the sequence data is not deposited in any database, it is speculative whether the resistance gene regions are identical. In another isolate from Belgium, the resistance genes *aadB*, *aphA1*, *strA*, and *tet*(B) were located on one contig but associated with a transposase of IS*Vsa3* and without information about the localization of the genes *sul2* and *floR*. Therefore, even if two *G. anatis* isolates contain the same antimicrobial resistance genes, their organization in the genome might differ, and further sequence analyses of *G. anatis* isolates are necessary to gain insight into the ability of *G. anatis* to include foreign DNA into its own genome.

It remains to be determined whether or not *G. anatis* plays a role in the pathogenesis of respiratory tract infections in cattle, especially calves. Nevertheless, *G. anatis* appears to be able to inhabit the bovine respiratory tract, can harbor a multitude of acquired antimicrobial resistance genes of Gram-positive and Gram-negative origin, and might act as an antimicrobial resistance gene reservoir for other respiratory pathogens. The finding of *G. anatis* isolates showing elevated MIC values for antimicrobial agents of seven classes and corresponding antimicrobial resistance genes or mutations in target structures in a fatal case of bovine respiratory tract infection in Germany, but also in neighboring countries, such as Belgium, is worrisome. The results of this and other studies [7,10,14] underline the request that antimicrobial agents should be applied to calves suffering from respiratory infections only after the identification of the causative pathogens and determination of their antimicrobial susceptibility profiles. Ongoing resistance monitoring of respiratory tract pathogens and detailed analysis of isolates displaying unusual antimicrobial susceptibility phenotypes is needed to identify resistance genotypes and the organization of the involved resistance genes to be able to assess the risk they may pose to animal health.

## 3. Materials and Methods

In May 2020, *Gallibacterium anatis* isolate P-279 (IMT49310) was identified in the state laboratory of Schleswig-Holstein, a federal state of Germany, during routine microbiology diagnostics of a lung sample taken at the post-mortem examination of an approximately 2-month old Angeln calf showing severe purulent and partially fibrinous pneumonia. The lung sample was streaked on MacConkey agar, Columbia agar, and BHI agar, the latter two supplemented with 5% (*v/v*) sheep blood and incubated for 18–24 h at 36 °C ± 0.5 °C under aerobic (MacConkey agar) and microaerophilic (Columbia and BHI agar) conditions. The obtained bacteria were identified to species level by matrix-assisted laser desorption/ionization time-of-flight mass spectrometry with Bruker Microflex LT in combination with Flex Control (flexControl Version 3.4) and BIOTYPER (MBT Compass 4.1) software (Bruker Daltonics, Bremen, Germany).

As no antimicrobial susceptibility testing (AST) standards exist for *G. anatis*, AST was performed according to the CLSI recommendations for *M. haemolytica.* For this, cation-adjusted Mueller–Hinton broth supplemented with 2.5% lysed horse blood served as the test medium. The inoculum was prepared with the colony suspension method using colonies from a sheep blood agar plate that was incubated for 18–24 h. The turbidity of the inoculum corresponded to a 0.5 McFarland standard. For broth microdilution, customized microtitre plates (Sensititre^TM^, Thermo Scientific, Waltham, MA, USA) were used, and broth macrodilution was performed for a single antimicrobial agent (kanamycin), not included in the microtitre plate layouts [11]. The microtitre plates were those also used in the National Resistance Monitoring program GE*RM*-Vet and contained 10–12 concentrations of the following antimicrobial agents (in mg/L) in two-fold dilution series: ampicillin (0.03–64), amoxicillin/clavulanic acid (0.03/0.015–64/32), penicillin G (0.015–32), ceftiofur (0.03–64), cefquinome (0.015–32), cephalothin (0.06–128), cefotaxime (0.015–32), cefoperazone (0.06–32), imipenem (0.015–32), oxacillin (0.015–8), neomycin (0.12–64), streptomycin (0.25–512), gentamicin (0.12–256), ciprofloxacin (0.008–16), enrofloxacin (0.008–16), marbofloxacin (0.008–16), nalidixic acid (0.06–128), tetracycline (0.12–256), doxycycline (0.06–128), erythromycin (0.015–32), tilmicosin (0.06–128), tylosin (0.06–128), tulathromycin (0.06–32), clindamycin (0.03–64), pirlimycin (0.03–64), florfenicol (0.12–256), trimethoprim/sulfamethoxazole (0.015/0.3–32/608), tiamulin (0.03–64), colistin (0.03–64), quinupristin/dalfopristin (0.015–32), linezolid (0.03–64), and vancomycin (0.015–32). The microtitre plates were incubated for 18–24 h at 35 °C ± 2 °C in ambient air. The following CLSI-recommended quality control strains were tested side-by-side with the *G. anatis* isolate: *Escherichia coli* ATCC^®^25922, *Streptococcus pneumoniae* ATCC^®^49619, and *M. haemolytica* ATCC^®^33396 (for ceftiofur and tulathromycin).

DNA was isolated from a pure culture with the MasterPure^TM^ Complete DNA and RNA Purification Kit (Epicentre, Madison, WI, USA); the quality was assessed using spectral analysis (NanoDrop Spectrophotometer, Thermo Fisher Scientific, Waltham, MA, USA), and the concentration determined with fluorimetrical quantification (Qubit 3.0 fluorometer double-stranded DNA broad-range assay kit, Thermo Fisher Scientific). Using the Library Preparation Kit Nextera XT (Illumina Inc., San Diego, CA, USA) and 1 ng of DNA, short-read sequencing was performed on a MiSeq^TM^ sequencer (MiSeq reagent Kit v3, Illumina Inc.), resulting in 300 bp paired-end reads. Long-read sequencing was performed on an Oxford Nanopore MinION (Oxford Nanopore Technologies, Oxford, UK), using 400 ng DNA to generate MinION one-dimensional libraries with the SQK-RBK004 kit, loaded onto an R4.9 flow cell. A closed genome was produced in a de novo hybrid assembly in which Unicycler v0.4.7 created a draft genome assembly with SPAdes v3.12, and used the MinION long-reads to connect the contigs. [15,16]. For the identification of acquired antimicrobial resistance genes and resistance mediating mutations, we used the software program ResFinder 4.1 (http://cge.food.dtu.dk/services/ResFinder/citations.php, accessed on 17 January 2023) [17]. In addition, the software programs Geneious^®^ 11.0.5, blastn and blastp (http://blast.ncbi.nlm.nih.gov/BLAST.cgi, accessed on 17 January 2023) and IS Finder (http://www-is.biotoul.fr, accessed on 17 January 2023) were used to analyze the sequence [18,19,20]. The nucleotide sequence has been deposited in the NCBI database under accession number CP110225.

## Figures and Tables

**Figure 1 antibiotics-12-00294-f001:**
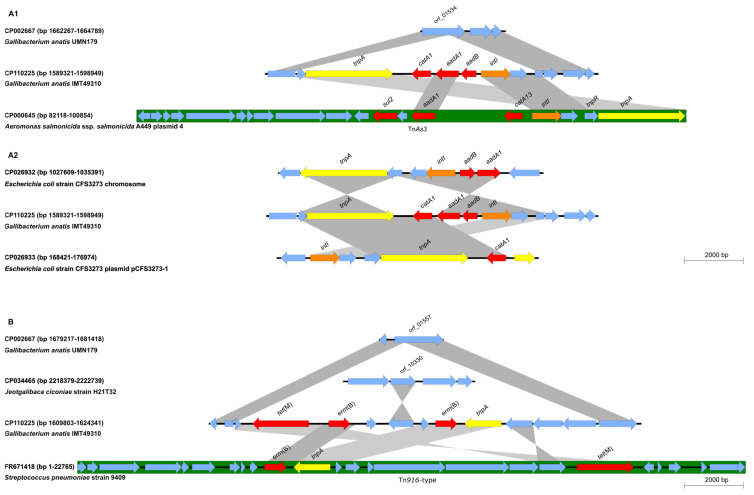
Organization of the resistance gene regions of *G. anatis* IMT49310. Open reading frames are shown as blue arrows, resistance genes are marked in red, transposase genes in yellow, and integrase genes in orange. The arrowheads indicate the direction of transcription. IS or integrative and conjugative elements and transposons are depicted as green boxes, and grey shading indicates similarities between sequences. (**A1**,**A2**) Schematic representation and alignment of resistance gene region I of *G. anatis* IMT49310 (CP110225) with sequences CP002667 (*G. anatis* UMN179), CP000645 (*Aeromonas salmonicida* ssp. *salmonicida* A449 plasmid 4), CP026932 (*Escherichia coli* strain CFS3273 chromosome) and CP026933 (*E. coli* strain CFS3273 plasmid pCFS3273-1) from GenBank. (**B**) Schematic representation and alignment of resistance gene region II of *G. anatis* IMT49310 (CP110225) with sequences CP002667 (*G. anatis* UMN179), CP034465 (*Jeotgalibaca ciconiae* strain H21T32 chromosome) and FR671418 (*Streptococcus pneumoniae* strain 9409 Tn*916*-type integrative and conjugative element) from Genbank. (**C1**,**C2**) Schematic representation and alignment of resistance gene region III of *G. anatis* IMT49310 (CP110225) with sequences CP002667 (*G. anatis* UMN179), CP029322 (*Pasteurella multocida* strain 14424), CP043925 (*Proteus columbae* strain T60 chromosome), and MW391932 (*Haemophilus parainfluenzae* strain HUB10329) from GenBank.

## Data Availability

The nucleotide sequence of *G. anatis* IMT49310 has been deposited in the NCBI database under accession number CP110225.

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
