# Peer review of "Genetic Organization of Acquired Antimicrobial Resistance Genes and Detection of Resistance-Mediating Mutations in a Gallibacterium anatis Isolate from a Calf Suffering from a Respiratory Tract Infection"

_antibiotics, 2023, doi:10.3390/antibiotics12020294_

Round 1

Reviewer 1 Report

Following are the suggestions.

1) Introduction need  more details regarding need of study, their statement of problem. These are missing in the introduction section.

2) Introduction and title must have some relation. The scheme of writing is not straight in introduction. 

3) Suggestion is to incorporate your results in tabular and figure form rather than text form. Your results are so confusing. Figure 1 must be break in different parts and write clearly.

4) Figure 1 is also not clear, their graphic is so blurred and color scheme is also not accordingly. 

5) On what basis, the author selected "As no antimicrobial susceptibility testing (AST) standards exist 203 for G. anatis, AST was performed according to the CLSI recommendations for M. haemo- 204 lytica"

6) Material and method is ok but needs more details for experimental works. 

Author Response

1) Introduction need  more details regarding need of study, their statement of problem. These are missing in the introduction section.

Answer:   As Gallibacterium anatis is not a common pathogen, we have briefly described the disease conditions from which G. anatis has been isolated in animals so far. We have also described that phenotypic resistance to various antimicrobial agents has been published, but that information on the genetic basis of antimicrobial resistance is largely missing. Hence we stated that the aim of our “study were to identify which acquired antimicrobial resistance genes and resistance-mediating mutations are present as well as to gain insight into their genetic organization in the genome of a G. anatis isolate from a German calf suffering from a respiratory tract infection.”

2) Introduction and title must have some relation. The scheme of writing is not straight in introduction. 

Answer:   We have modified the title a bit so that it corresponds better to the content of the manuscript, including the Introduction.

3) Suggestion is to incorporate your results in tabular and figure form rather than text form. Your results are so confusing. Figure 1 must be break in different parts and write clearly.

Answer:   Figure 1 has been modified and is now presented in a larger format on two pages. Moreover, we have carefully re-checked the text and included some changes to enhance clarity. We also tried to transform the text into a table, but were rather unsatisfied with the result. Thus, we improved the description in the text and deleted the table again.

4) Figure 1 is also not clear, their graphic is so blurred and color scheme is also not accordingly. 

Answer:   The resolution of the figure corresponds exactly to the requirements of the journal. Nevertheless, we agree that Figure 1 is a bit small and the details are hard to read. Thus, we have followed the reviewer’s advice, have changed the format and spread this figure over two pages. Now, everything is much better readable. We checked the color coding again and could not find any inconsistencies.

5) On what basis, the author selected "As no antimicrobial susceptibility testing (AST) standards exist 203 for G. anatis, AST was performed according to the CLSI recommendations for M. haemo- 204 lytica"

Answer:   G. anatis resembles closely M. haemolytica in its growth characteristics and both organisms are found in the respiratory tract of cattle. As there are currently no approved AST standards that state how AST of G. anatis should be performed, the next best option is to perform AST of G. anatis according to the method approved by CLSI for M. haemolytica and described in the VET01S document. The Belgian study (Ref 7) also followed the VET01S document for AST although they did not mention exactly which protocol the used.

However, the fact that the phenotypic AST results corresponded very well with the resistance genes and resistance-mediating mutations detected in our G. anatis isolate suggests that the phenotypic AST method applied might be suitable.

6) Material and method is ok but needs more details for experimental works. 

Answer:   We have added more details for the AST method applied and the quality controls incuded.

Reviewer 2 Report

Dear Editor

This brief report is about the investigation of the antimicrobial susceptibility of a Gallibacterium annatis strain isolated from a calf. The manuscript is well written and the topic is about a rather interesting, emerging pathogen. The results are interesting and elucidate the possibility of acquired resistance of this pathogen and perhaps its role on the spread of resistance to other bacteria. The only point I would like to make is that the images are of low quality and therefore should be provided in a clearer format, a review that does not need my further approval. Other than that this is a fine manuscript for publication in Antibiotics.

Author Response

This brief report is about the investigation of the antimicrobial susceptibility of a Gallibacterium annatis strain isolated from a calf. The manuscript is well written and the topic is about a rather interesting, emerging pathogen. The results are interesting and elucidate the possibility of acquired resistance of this pathogen and perhaps its role on the spread of resistance to other bacteria. The only point I would like to make is that the images are of low quality and therefore should be provided in a clearer format, a review that does not need my further approval. Other than that this is a fine manuscript for publication in Antibiotics.

Answer:   We thank the reviewer for the overall positive comments. Concerning Figure 1, the resolution of the figure corresponds exactly to the requirements of the journal. Nevertheless, we agree that Figure 1 is a bit small and the details are hard to read. Thus, we have followed the reviewer’s advice, have changed the format and spread this figure over two pages. Now, everything is much better readable.

Reviewer 3 Report

Dear Authors,

Many thanks for submitting your exciting work for revision. Very interesting study aimed to analyse the genetic organization of acquired resistance genes in Gallibacterium anatis isolate IMT49310, which was obtained from a German calf suffering from a respiratory tract infection.

You cannot really make any judgement based on only one sample of Gallibacterium anatis

 Please have a look at the comments.  With more work, this could be a great paper.

·         Keywords – I suggest you use additional words that are different from those in the title. This will increase the chance of your work being found by search engines. For example, antimicrobial resistance; cattle; multidrug-resistant; respiratory diseases; whole-genome sequencing; zoonoses.

·         The introduction does not have sufficient information regards the significance of this type of bacteria and the zoonotic effect.

·         Can you please ensure to define the abbreviation of Gallibacterium anatis before you use it? 

·         Too small a sample size

·         No details regarding the animal's ethics.

·         The materials and methods must be described in detail. Sufficient information is missing

·         There was no explanation for why the authors selected only one strain of bacterial

·         No information regarding the calves’ history (age, breed) and any other pathogens detected?

·         No information was provided regarding the bacterial culture of the sample and the sample identification.

·         No details regarding the antimicrobial susceptibility testing and how it was performed by disc diffusion as per Clinical and Laboratory Standards Institute guidelines for the selected antimicrobials.

·         There was no mention of breaking point references for antimicrobial resistance.

·         Furthermore, no mention of the concentration of the selected antimicrobials.

·         No details methods were provided regarding the antimicrobial-susceptibility testing.

·         There is no description of how the authors performed genotypic resistance gene detection.

·         No sufficient information on the methods regarding the antimicrobial MICs performed.

·         The methods need more detail about statistical analysis.

·         You should emphasise that the sample size is really small in the discussion

·         The figures are very difficult to read and follow.

·         The take-home messages need to be much more sophisticated, practical and focused on the findings of this specific manuscript.

·         The discussion section is a bit superficial and doesn’t bring in the study’s findings to meaningfully recommend how these results should lead to improved practices.

·         References used do not have enough impact factors and/or are outdated

Author Response

You cannot really make any judgement based on only one sample of Gallibacterium anatis

Please have a look at the comments.  With more work, this could be a great paper.

Answer:   We feel that there is a misunderstanding. We did not make any extrapolations from the results obtained with this single isolate. However, this is - to the best of our knowledge - the first multiresistant G. anatis isolate obtained from a respiratory tract infection of a calf in Germany. Unfortunately, there are not more bovine G. anatis isolates available to us. If so, we surely would have included them in this study. Nevertheless, the detailed molecular analysis of this single isolate provides relevant information on the organization and genetic contexts of acquired antimicrobial resistance genes and the detection of resistance-mediating mutations present in this isolate. This information will be of relevance for comparisons if in the future more G. anatis isolates with similar characteristics should be found.  

  • Keywords – I suggest you use additional words that are different from those in the title. This will increase the chance of your work being found by search engines. For example, antimicrobial resistance; cattle; multidrug-resistant; respiratory diseases; whole-genome sequencing; zoonoses.

Answer:   We have followed the reviewer’s advice and incorporated new key words: antimicrobial resistance; cattle; multidrug resistance; respiratory disease; whole-genome sequencing

  • The introduction does not have sufficient information regards the significance of this type of bacteria and the zoonotic effect.

Answer:   Whether or not, G. anatis is a zoonotic pathogen needs to be determined. To the best of our knowledge, there is a single report that described a human bacteremia case due to G. anatis. In this particular case, the affected human was a highly immunocompromised patient. Thus, based on the current knowledge, we have not further explored the zoonotic aspects of G. anatis.

As G. anatis is not a common pathogen (at least not yet in cattle), we have briefly described in the introduction the disease conditions from which G. anatis has been isolated in animals so far. We have also described that phenotypic resistance to various antimicrobial agents has been published, but that information on the genetic basis of antimicrobial resistance - especially data on the organization of resistance genes - is largely missing in G. anatis.

Furthermore, we have clearly defined the aims of our study and stated that the aims of our “study were to identify which acquired antimicrobial resistance genes and resistance-mediating mutations are present as well as to gain insight into their genetic organization in the genome of a G. anatis isolate from a German calf suffering from a respiratory tract infection.”

  • Can you please ensure to define the abbreviation of Gallibacterium anatis before you use it?

Answer:   Thanks a lot for this advice. We have now defined the abbreviation at first use in the Abstract and the text.

  • Too small a sample size

Answer:   As mentioned above, this is - to the best of our knowledge - the first multiresistant G. anatis isolate obtained from a respiratory tract infection of a calf in Germany. There are not more bovine G. anatis isolates currently available to us.

  • No details regarding the animal's ethics.

Answer:   As this isolate originated from a case of fatal respiratory disease and the lung sample (from which the isolate was obtained) was taken during necropsy of the calf, there is no need to obtain approval from an ethics committee.

  • The materials and methods must be described in detail. Sufficient information is missing

Answer:   We have expanded the Materials and Methods part.

  • There was no explanation for why the authors selected only one strain of bacterial

Answer:   We had only this one lung sample from the deceased calf and all the isolates obtained from this sample were phenotypically indistinguishable. Moreover, antibiograms done for three isolates from the same sample yielded the same results, suggesting that the calf was infected by a pure culture of G. anatis.

  • No information regarding the calves’ history (age, breed) and any other pathogens detected?

Answer:   The diagnostic laboratory, from which we received the G. anatis sample, confirmed that the deceased animal was an approximately 2-month old Anglen calf. Unfortunately, no history of antimicrobial treatment was available. The lung sample yielded a pure culture of G. anatis.

  • No information was provided regarding the bacterial culture of the sample and the sample identification.

Answer:   We added the information concerning the bacterial culture and identification: “The lung sample was streaked on MacConkey agar, Columbia agar and BHI agar, the latter two supplemented with 5% (v/v) sheep blood and incubated for 18-24 h at 36°C +/- 0.5°C under aerobic (MacConkey agar) and microaerophilic (Columbia and BHI agar) conditions. The obtained bacteria were identified to species level by matrix-assisted laser desorption/ionization time-of-flight mass spectrometry with Bruker Microflex LT in com-bination with Flex Control (flexControl Version 3.4) and BIOTYPER (MBT Compass 4.1) software (Bruker Daltonics, Bremen, Germany).”

  • No details regarding the antimicrobial susceptibility testing and how it was performed by disc diffusion as per Clinical and Laboratory Standards Institute guidelines for the selected antimicrobials.

Answer:   We have added more details for the AST method applied and the quality controls incuded. We have not used disk diffusion, but broth microdilution for most of the tested antimicrobial agents. Solely for kanamycin, which was not included in the microtitre plates used, we performed broth macrodilution.

  • There was no mention of breaking point references for antimicrobial resistance.

Answer:   This is true, as we tried to avoid a classification of the isolate as resistant - intermediate - susceptible. This was due to the fact that there are no approved breakpoints applicable to G. anatis (as explained in the first sentence of the results section). As a consequence, we have just listed in the first part of the results section the antimicrobial agents for which elevated minimal inhibitory concentrations were recorded.

  • Furthermore, no mention of the concentration of the selected antimicrobials.

Answer:   For the AST methods applied, there is not a single concentration that was tested. All selected antimicrobial agents, were tested in 10-12 different concentrations in two-fold dilution series. This information has been added to Materials & Methods.

  • No details methods were provided regarding the antimicrobial-susceptibility testing.

Answer:   Details have been added. Please see the revised section of the text.

  • There is no description of how the authors performed genotypic resistance gene detection.

Answer:   This is not entirely correct. We have indicated that we have used the rResFinder 4.1 (http://cge.food.dtu.dk/services/ResFinder/citations.php) software. Now we have explicitly stated “For the identification of acquired antimicrobial resistance genes and resistance mediating mutations, we used the software program rResFinder 4.1 (http://cge.food.dtu.dk/services/ResFinder/citations.php) [17].”

  • No sufficient information on the methods regarding the antimicrobial MICs performed.

Answer:   This is not entirely correct. We have mentioned that broth microdilution and broth macrodilution have been used as AST methods according to the recommendations of the CLSI and have cited the respective CLSI document VET01S. As the VET01S document is freely available, we thought that it is not relevant to mention all the details of the AST procedure. Now, we have added the relevant details.

  • The methods need more detail about statistical analysis.

Answer:   This is a purely descriptive study on a single isolate. After having talked to our statistics department, the experts confirmed that for this study statistical analyses are not needed.

  • You should emphasise that the sample size is really small in the discussion

Answer:   The entire manuscript is about the findings obtained from a single isolate. This has already been mentioned at various places in the text.

  • The figures are very difficult to read and follow.

Answer:   The resolution of the figure corresponds exactly to the requirements of the journal. Nevertheless, we agree that Figure 1 is a bit small and the details are hard to read. Thus, we have followed the reviewer’s advice, have changed the format and spread this figure over two pages. Now, everything is much better readable.

  • The take-home messages need to be much more sophisticated, practical and focused on the findings of this specific manuscript.

Answer:   The final paragraph of the Results and Discussion section summarises the findings of the present study in the context of other studies and provides an outlook into what is needed in the future. Moreover, we have added another sentence to provide an additional take home message.

  • The discussion section is a bit superficial and doesn’t bring in the study’s findings to meaningfully recommend how these results should lead to improved practices.

Answer:   We have added a sentence to put the findings of this study in context with recommendations for improved application of antimicrobial agents.

  • References used do not have enough impact factors and/or are outdated

Answer:   We were a bit surprised to read that the references cited have not have enough impact factors and/or are outdated as most references are from 2018-2022. We have checked the literature again and did not find better references. It would have been helpful if the reviewer had told us which of the apparently outdated references should be replaced by more actual ones? By the way, all other three reviewers stated that all the cited references are relevant to the research.

Reviewer 4 Report

This manuscript presents some interesting aspects on Gallibacterium anatis isolate and related antimicrobial resistance.

Author Response

This manuscript presents some interesting aspects on Gallibacterium anatis isolate and related antimicrobial resistance.

Answer:   We thank the reviewer for the positive evaluation.

Round 2

Reviewer 1 Report

After modification, it sounds good.

Reviewer 3 Report

Dear Authors, 

Thanks for addressing the feedback. The manuscript, particularly the Materials and Methods sections, is much improved. I propose that the work be accepted in its existing form. 

Kind Regards